# Sit4 Genetically Interacts with Vps27 to Regulate Mitochondrial Function and Lifespan in *Saccharomyces cerevisiae*

**DOI:** 10.3390/cells13080655

**Published:** 2024-04-09

**Authors:** Telma S. Martins, Miguel Correia, Denise Pinheiro, Carolina Lemos, Marta Vaz Mendes, Clara Pereira, Vítor Costa

**Affiliations:** 1i3S—Instituto de Investigação e Inovação em Saúde, Universidade do Porto, 4200-135 Porto, Portugal; telma.martins@ibmc.up.pt (T.S.M.); miguelc@i3s.up.pt (M.C.); dpinheiro@i3s.up.pt (D.P.); clclemos@ibmc.up.pt (C.L.); mvm@ibmc.up.pt (M.V.M.); 2IBMC—Instituto de Biologia Molecular e Celular, Universidade do Porto, 4200-135 Porto, Portugal; 3ICBAS—Instituto de Ciências Biomédicas Abel Salazar, Universidade do Porto, 4050-313 Porto, Portugal

**Keywords:** Sit4, Vps27, vacuolar trafficking, mitochondria, iron, chronological aging

## Abstract

The Sit4 protein phosphatase plays a key role in orchestrating various cellular processes essential for maintaining cell viability during aging. We have previously shown that *SIT4* deletion promotes vacuolar acidification, mitochondrial derepression, and oxidative stress resistance, increasing yeast chronological lifespan. In this study, we performed a proteomic analysis of isolated vacuoles and yeast genetic interaction analysis to unravel how Sit4 influences vacuolar and mitochondrial function. By employing high-resolution mass spectrometry, we show that *sit4*Δ vacuolar membranes were enriched in Vps27 and Hse1, two proteins that are part of the endosomal sorting complex required for transport-0. In addition, *SIT4* exhibited a negative genetic interaction with *VPS27*, as *sit4*∆*vps27*∆ double mutants had a shortened lifespan compared to *sit4*∆ and *vps27*∆ single mutants. Our results also show that Vps27 did not increase *sit4*∆ lifespan by improving protein trafficking or vacuolar sorting pathways. However, Vps27 was critical for iron homeostasis and mitochondrial function in *sit4*∆ cells, as *sit4*∆*vps27*∆ double mutants exhibited high iron levels and impaired mitochondrial respiration. These findings show, for the first time, cross-talk between Sit4 and Vps27, providing new insights into the mechanisms governing chronological lifespan.

## 1. Introduction

Lysosomes and lysosome-like vacuoles in yeast serve as pivotal hubs for maintaining cellular homeostasis, metabolism, and lifespan. These organelles are enriched in carrier proteins and hydrolases, such as proteases, glycosidases, lipases, nucleases, phosphatases, and sulfatases. The concentration of these hydrolases and the vacuolar H^+^-ATPase-dependent maintenance of an acidic pH render lysosomes a major catabolic center [1,2,3,4,5]. Notably, lysosomal dysfunction is a hallmark of aging common to several age-related diseases, including cancer and neurodegenerative and metabolic disorders [1,6]. The decline in lysosomal function during aging has been linked to the dysregulation of nutrient signaling pathways, an increase in lysosomal pH, and a decrease in autophagy [1,7,8,9,10,11]. Notably, lifespan extension by dietary restriction involves the modulation of nutrient signaling pathways, which in turn depend at least in part on lysosome function in several organisms [7,12,13,14,15,16,17,18].

Lysosomes degrade both intracellular and exogenous components, from macromolecules to organelles, such as mitochondria, peroxisomes, or lipid droplets, and even surface receptors and pathogens. The major pathways of substrate delivery to lysosomes are autophagy and the endocytic pathway [2,19]. For example, during macroautophagy (hereafter named autophagy), a double membrane is formed around cytoplasmic material targeted for degradation, leading to the generation of autophagosomes that ultimately merge with lysosomes [20,21]. Conversely, mitophagy allows the selective clearance of mitochondria [22]. Autophagy is highly conserved during evolution and serves critical roles in ensuring cell survival [23,24]. It maintains protein and organelle quality control by removing damaged material that is generated under stress conditions and during aging or diseases. In addition, it allows cells to survive under nutrient deprivation by recycling macromolecules, to provide amino acids, sugars, fatty acids, nucleotides, and other nutrients. The endocytic pathway plays a vital role in regulating cellular functions and immune responses by delivering extracellular and cell surface cargos to lysosomes via endocytosis [25,26]. Internalized proteins localized at early endosomes undergo sorting, being either targeted for degradation in lysosomes or recycled to other compartments (e.g., the Golgi apparatus and the plasma membrane). Proteins destined for degradation are ubiquitinated, allowing their recognition by the endosomal sorting complex required for transport (ESCRT), which facilitates its sorting via the multivesicular body (MVB) pathway to the lysosome. Conversely, proteins that are not ubiquitinated follow the recycling pathway [27,28].

Trafficking pathways play a vital role in the function and biogenesis of vacuoles. In eukaryotic cells, organellar proteins are produced in the ER, moved to the Golgi apparatus, and subsequently sorted into their cellular compartments. In yeast, proteins destined for the vacuole can follow three routes: the carboxypeptidase Y (CPY) pathway (indirectly, through late endosomes/MVB), the alkaline phosphatase (ALP/AP-3) pathway [26,29], or the cytoplasm to vacuole targeting (Cvt) pathway, which uses much of the autophagic machinery to transport hydrolases, such as α-mannosidase (Ams1) and aminopeptidase I (Ape1), from the cytoplasm to the vacuole [30].

Apart from their essential function in degradation, lysosomes also serve as storage site for metabolites derived from the diet (e.g., amino acids, sugars, lipids, nucleotides, and ions) and play a central role in regulating nutrient sensing and signaling pathways [4,5]. For instance, vacuolar iron storage is particularly important to prevent oxidative stress, as it is a redox-active metal [31].

Targeting lysosomes to optimize their function is emerging as a means to promote a healthier lifespan [1,3,6,7,11]. This promising strategy requires a deeper comprehension of the molecular cues driving the decline in lysosomal function with age and its improvement under lifespan extension conditions. We have previously shown that *SIT4* deletion increases yeast chronological lifespan (CLS) by promoting vacuolar acidification as well as mitochondrial derepression and oxidative stress resistance [32,33,34,35]. Sit4 is a PP2A-like Ser/Thr protein phosphatase, a homologue of the mammalian protein phosphatase 6 (PP6) [36], which is also involved in the regulation of the cell cycle and budding [37,38], cell wall integrity, actin cytoskeleton organization and ribosomal gene expression mediated by Pkc1 function [39], nutrient signaling [40,41], carbohydrate and lipid metabolism [42,43], pH and monovalent ion homeostasis [44], and ER-to-Golgi trafficking [45]. Sit4 acts as the catalytic subunit of several protein complexes, formed by its association with distinct regulatory subunits, explaining its role across a broad array of cellular functions. For instance, Sit4-associated proteins (SAPs), namely Sap4, Sap155, Sap185, or Sap190, bind Sit4 independently to positively regulate cell cycle progression at the G1 phase [46]. Sit4 can also form a ceramide-activated protein phosphatase complex by interacting with regulatory subunits Tpd3 and Cdc55 [47]. In addition, Sit4 interacts with and is inhibited by Tap42 in response to nutrient signals through TORC1 [40]. The dissociation of the Tap42-Sit4 complex results in Sit4 activation and dephosphorylation of the GATA-binding transcription factor Gln3, which is then translocated into the nucleus where it activates the nitrogen catabolite repression pathway [48].

In this study, we assessed how Sit4 deficiency impacts vacuoles by performing a proteomic analysis of vacuolar membranes from wild-type and *sit4*∆ cells and investigated the role of Vps27 in regulating *sit4*∆ CLS. Our findings reveal that Vps27 is enriched in the vacuolar membranes of *sit4*∆ cells and is crucial for lifespan extension by modulating mitochondrial function.

## 2. Materials and Methods

### 2.1. Yeast Strains and Growth Conditions

The *S. cerevisiae* strains utilized in this study are detailed in Table 1. Yeast cells were cultured aerobically at 26 °C on an orbital shaker set at 140 rpm, maintaining a flask volume to medium volume of 5:1. Synthetic complete (SC) medium was used as a growth medium, comprising drop-out, 2% (*w*/*v*) glucose (Thermo Fisher Scientific, Waltham, MA, USA), and 0.67% (*w*/*v*) yeast nitrogen base without amino acids (BD BioSciences, San Jose, CA, USA), supplemented with the required amino acids and nucleotides. For BY4741 strains, SC medium was supplemented with histidine (0.008% (*w*/*v*), Sigma Aldrich, St. Louis, MO, USA), methionine (0.038% (*w*/*v*), Sigma Aldrich, St. Louis, MO, USA), leucine (0.04% (*w*/*v*), Sigma Aldrich, St. Louis, MO, USA), and uracil (0.008% (*w*/*v*), Sigma Aldrich, St. Louis, MO, USA). For BY4742 strains, SC medium was supplemented with histidine (0.008% (*w*/*v*), Sigma Aldrich, St. Louis, MO, USA), lysine (0.008% (*w*/*v*), Sigma Aldrich, St. Louis, MO, USA), leucine (0.04% (*w*/*v*), Sigma Aldrich, St. Louis, MO, USA), and uracil (0.008% (*w*/*v*), Sigma Aldrich, St. Louis, MO, USA). To assess Mup1-GFP levels in response to methionine, BY4742 cells were cultured in minimal medium (MM), containing 2% (*w*/*v*) glucose (Thermo Fisher Scientific, Waltham, MA, USA) and 0.67% (*w*/*v*) yeast nitrogen base without amino acids (BD BioSciences, San Jose, CA, USA), supplemented with amino acids or nucleotides [0.008% (*w*/*v*) histidine (Sigma Aldrich, St. Louis, MO, USA), 0.008% (*w*/*v*) lysine (Sigma Aldrich, St. Louis, MO, USA), 0.04% (*w*/*v*) leucine (Sigma Aldrich, St. Louis, MO, USA), and 0.008% (*w*/*v*) uracil (Sigma Aldrich, St. Louis, MO, USA)]. Furthermore, 38 mg L^−1^ of L-methionine (Thermo Fisher Scientific, Waltham, MA, USA) was added to cell cultures.

To assess the growth in respiratory medium, cells were grown in YPD [1% (*w*/*v*) yeast extract (Liofilchem, Roseto degli Abruzzi, Italy), 2% (*w*/*v*) peptone (Liofilchem, Roseto degli Abruzzi, Italy), and 2% (*w*/*v*) glucose (Thermo Fisher Scientific, Waltham, MA, USA)] or YPG [1% (*w*/*v*) yeast extract (Liofilchem, Roseto degli Abruzzi, Italy), 2% (*w*/*v*) peptone (Liofilchem, Roseto degli Abruzzi, Italy), and 3% (*v*/*v*) glycerol (Thermo Fisher Scientific, Waltham, MA, USA)] supplemented with 1.5% (*w*/*v*) agar (Liofilchem, Roseto degli Abruzzi, Italy).

The deletion of *VPS27* was carried out using a deletion fragment encompassing *KanMx4* or *MXHIS3* and *VPS27* flanking regions. Yeast cells were transformed by the lithium acetate/single-stranded carrier DNA/PEG method [51], and gene deletion was validated using standard PCR procedures.

### 2.2. Isolation of Vacuolar Membranes and Proteomic Analysis by HPLC-MS/MS

Cells were cultured to the late exponential (Log) phase (OD_600nm_ ≈ 2) and vacuolar membranes were isolated as described [52,53]. Samples were processed for proteomics analysis and proteins were identified and quantified by mass spectrometry (nanoLC-MS/MS), as described [54,55]. Proteomic datasets were analyzed using Proteome Discoverer 2.5.0.400 software (Thermo Fisher Scientific, Waltham, MA, USA), the UniProt database for the *S. cerevisiae* Proteome 2020_03, and a contaminant database from MaxQuant (version 1.6.2.6, Max Planck Institute of Biochemistry, Munich, Germany). The levels of peptides were normalized based on total peptide levels of annotated vacuolar membrane proteins.

To assess group differences in protein levels, the homogeneity of variances was evaluated using the Levene test, and pairwise comparisons were conducted using Student’s *t*-test. Statistical analysis was carried out using the Statistical Package for the Social Sciences (SPSS), version 26 (IBM Corp., Armonk, NY, USA), with the significance levels set at α = 0.05. For the statistical analysis of the overrepresentation of functional groups, the Gene Ontology (GO) database (6 March 2023 and 10.5281/zenodo.7709866) was utilized.

### 2.3. Chronological Lifespan

CLS was evaluated as described [56]. To study the effect of iron chelation on CLS, the growth medium was supplemented with 80 µM bathophenanthrolinedisulfonate (BPS; disodium salt hydrate; Sigma Aldrich, St. Louis, MO, USA). Cells were grown overnight in SC medium, supplemented or not with BPS, diluted to an OD_600nm_ = 0.2 in SC medium and further grown for 24 h (considered time zero, t0, in the lifespan assay). These cultures were kept at 26 °C, and cell viability was assessed over time using standard dilution plate counts on YPD medium supplemented with 1.5% (*w*/*v*) agar (Liofilchem, Roseto degli Abruzzi, Italy). Following incubation at 26 °C for 2–3 days, colonies were counted and the percentage of the colony-forming units (CFUs) relative to t0 was determined.

### 2.4. Glucose and Ethanol Determination

Cells were grown in SC medium, and samples from the extracellular medium were taken at 12, 15, 24, and 48 h of growth after the Log phase. Glucose levels were quantified using the D-Glucose GOD-POD kit (Nzytech, Lisbon, Portugal) following the method described in the manufacturer’s instructions. For ethanol determination, samples were diluted 100-fold with H_2_O, and ethanol levels were quantified using the Enzytec Liquid Ethanol kit (r-biopharm, Darmstadt, Germany) following the method described in the manufacturer’s instructions.

### 2.5. Glycogen Measurement

Cells were grown for 3 days in YPD plates and glycogen accumulation was qualitatively measured by exposing yeast colonies to iodine crystals. Yeast colonies develop a brown coloration that reflects their glycogen content [57].

### 2.6. Western Blotting

Cells were collected at the Log (OD_600nm_ ≈ 0.6), late Log (OD_600nm_ ≈ 2), and post-diauxic shift (PDS; 24 h after Log) phases, as indicated in figure legends. For the analysis of microautophagy induction, Log phase cells treated for 4 h with 200 ng mL^−1^ of rapamycin were used as positive control. Proteins were extracted using 0.1 M NaOH (Merck, Darmstadt, Germany), dissolved in Laemmli buffer, and quantified using the Pierce BCA Protein Assay Kit (Thermo Fisher Scientific, Waltham, MA, USA), as described [53]. Protein samples were prepared by adding 5% (*v*/*v*) 2-mercaptoethanol (Merck, Darmstadt, Germany), separated by SDS-PAGE, and transferred onto a nitrocellulose membrane (GE Healthcare, Chicago, IL, USA). For Mup1-GFP detection, cells were vortexed in Laemmli buffer in the presence of zirconium beads and then incubated at 60 °C for 10 min. To assess Mup1-GFP levels in response to methionine, proteins were extracted as previously reported [58]. Cell lysis was performed by incubating the cell pellets in 0.5 mL of 0.2 M NaOH (Merck, Darmstadt, Germany), 0.2% (*v*/*v*) 2-mercaptoethanol (Merck, Darmstadt, Germany) for 10 min on ice and adding 5% (*w*/*v*) trichloroacetic acid (Thermo Fisher Scientific, Waltham, MA, USA), followed by a 10 min incubation on ice. The pellet was collected by centrifugation, washed twice with acetone (Vidrolab 2, Gandra, Portugal), suspended in Laemmli buffer plus zirconium beads, vortexed, and then incubated for 10 min at 37 °C.

Immunodetection was performed using mouse anti-HA (1:1000; Santa Cruz Biotecnology, Dallas, TX, USA, sc-7392), mouse anti-GFP (1:5000; Roche, Basel, Switzerland, 11814460001), mouse anti-CPY (1:2000; Invitrogen, Waltham, MA, USA, A-6428), goat anti-ApeI (1:200; Santa Cruz Biotechnology, Dallas, TX, USA, sc-26740), mouse anti-Por1 (1:5000; Invitrogen, Waltham, MA, USA, 459500), rabbit anti-Atp2 (1:2000; Abcam, Cambridge, UK, ab128743), or mouse anti-Cox2 (1:7000; Invitrogen, Waltham, MA, USA, 4B12A5) as primary antibodies, and anti-rabbit IgG-peroxidase (1:5000; Sigma Aldrich, St. Louis, MO, USA), anti-mouse IgG-peroxidase (1:5000; Molecular Probes, Eugene, OR, USA), or anti-goat IgG-peroxidase (1:5000; Sigma Aldrich, St. Louis, MO, USA) as secondary antibodies. The primary antibodies used to probe the loading controls were mouse anti-Hxk2 (1:100,000, Rockland, Philadelphia, PA, USA, 200-4159) or mouse anti-GAPDH (1:30,000, Proteintech, Sankt Leon-Rot, Germany, 60004-1). The secondary antibody was anti-mouse IgG-peroxidase (1:5000, Molecular Probes, Eugene, OR, USA). Immunodetection was performed by chemiluminescence (ECL, Advansta, San Jose, CA, USA).

### 2.7. Colony Immunoblot Assay

Yeast cultures were diluted to OD_600nm_ of 0.15 using phosphate-buffered saline (PBS, 137 mM NaCl (Thermo Fisher Scientific, Waltham, MA, USA), 2.7 mM KCl (Merck, Darmstadt, Germany), 10 mM Na_2_HPO_4_ (Panreac, Barcelona, Spain), 1.8 mM KH_2_PO_4_ (Panreac, Barcelona, Spain), pH 7.4), and 13.3 μL of culture was spotted onto a SC glucose plate. Upon 72 h of incubation at 26 °C, colonies were covered with a nitrocellulose membrane (GE Healthcare, Chicago, IL, USA). After 24 h at 26 °C, membranes were removed and washed with TBS-0.05% (*v*/*v*) Tween (Sigma Aldrich, St. Louis, MO, USA) solution to clear away all cells that remained attached. Detection of CPY secretion was carried out by immunoblotting, with mouse anti-CPY primary antibody (1:2000, Invitrogen, Waltham, MA, USA) and anti-mouse IgG-peroxidase secondary antibody (1:5000, Molecular Probes, Eugene, OR, USA).

### 2.8. Iron Levels

Total iron levels were quantified in yeast cells (6 × 10^8^ cells mL^−1^) grown in SC glucose medium to late Log (OD_600nm_ ≈ 2) and PDS phases, using a colorimetric assay, as previously described [59].

### 2.9. Oxidative Stress Resistance

For the analysis of oxidative stress resistance, cells were grown to Log phase in SC medium supplemented or not with 80 µM BPS, and then treated with 1.5 mM hydrogen peroxide (H_2_O_2_; Merck, Darmstadt, Germany) for 1 h. Cell viability was determined as described above.

### 2.10. Growth Analysis

Respiratory capacity was assessed by evaluating cell growth in glycerol medium. Yeast cultures were diluted to OD_600nm_ = 0.1, and four ten-fold serial dilutions were prepared in PBS. Then cells were spotted onto YPD (glucose) or YPG (glycerol) containing 1.5% (*w*/*v*) agar (Liofilchem, Roseto degli Abruzzi, Italy), and the plates were incubated at 26 °C for 2–3 days. To evaluate iron sensitivity, four ten-fold serial dilutions of each culture were spotted onto SC plates supplemented or not with iron(II) sulfate heptahydrate (10 mM, 20 mM or 25 mM; Sigma Aldrich, St. Louis, MO, USA), previously dissolved in citrate buffer [81.9 mM sodium citrate (Sigma Aldrich, St. Louis, MO, USA), 18.1 mM Citric acid (Merck, Darmstadt, Germany)], and the plates were incubated for 3 days at 26 °C.

### 2.11. Oxygen Consumption

Oxygen consumption rate (OCR) was measured using a Clark-type oxygen electrode coupled to an Oxygraph plus system (Hansatech, Norfolk, UK), as described [53].

### 2.12. Alkaline Phosphatase Assay

For the analysis of alkaline phosphatase activity, protein extracts were prepared and incubated with nitrophenylphosphate (Sigma Fast, Sigma Aldrich, St. Louis, MO, USA), as described [49].

### 2.13. Mitochondrial Mass

The mitochondrial mass was evaluated in cells expressing GFP fused to the mitochondrial pre-sequence preSU9 (pVT100U-*preSU9-GFP*), which has low sensitivity to the mitochondrial membrane potential [60,61]. Cells were grown to Log phase, and GFP fluorescence was assessed by flow cytometry using a BD Accuri C6 (20,000 events on the FL1 channel) and the FlowJo v10.8.1 software version (BD Life Sciences, Franklin Lakes, NJ, USA).

### 2.14. Analysis of Mitochondrial DNA by qPCR

Cells were grown to Log phase in SC medium, and total DNA isolated using the GeneJET Genomic DNA Purification kit (Thermo Fisher Scientific, Waltham, MA, USA), following the method described in the manufacturer’s instructions. The concentration and purity of all DNA stock solutions were determined with a NanoDrop ND-1000 spectrophotometer (Thermo Fisher Scientific, Waltham, MA, USA) and DNA integrity checked on agarose gels. For each sample, the total DNA concentration was normalized to 5 ng/µL. For qPCR amplifications, 2 µL of template DNA (dilution 1/250) was used with the primer pairs (0.25 µM of each primer) listed in Table 2 and 10 µL of iTaq Universal SYBR Green supermix (Bio-Rad, Hercules, CA, USA). qPCR tests were performed using an iCycler iQ5 real-time PCR detection system (Bio-Rad, Hercules, CA, USA) with the following settings: 95 °C for 3 min; 45 cycles of 95 °C for 15 s, 55 °C for 30 s and 60 °C for 30 s. The relative efficiency and quality of each primer pair was assessed using a 5-fold serial dilution series of the DNA (1/10, 1/50, 1/250, 1/1250, and 1/6250). Negative controls (non-template controls) were included in all qPCR tests. To exclude the formation of nonspecific products, a melting curve analysis was performed at the end of each qPCR. qPCR analysis included two independent biological replicates and four technical replicates for each DNA sample. The data obtained were analyzed using the method described by Pfaffl [62] and the CFX Maestro 2.0 (v 5.0.021.0616) software (Bio-Rad, Hercules, CA, USA), to obtain the total levels of mitochondrial DNA (mtDNA) relative to nuclear DNA in the tested strains. *ACT1* was used as a reference for nuclear DNA, and *COX1* and *COX3* were used as references for mtDNA.

### 2.15. Statistical Analysis

Data were analyzed using GraphPad Prism Software v10.0.0 (GraphPad Software, Boston, MA, USA). Statistical comparisons were performed using one-way ANOVA, two-way ANOVA, or Student’s *t*-test as appropriate. Significance levels were represented as follows: *, *p* ≤ 0.05; **, *p* ≤ 0.01; ***, *p* ≤ 0.001; ****, *p* ≤ 0.0001.

## 3. Results

### 3.1. Vps27 Is Enriched in the Vacuolar Membranes and Mediates the Lifespan Extension of Cells Lacking the Phosphatase Sit4

To investigate the impact of Sit4 deficiency on yeast lysosome-like vacuoles, we conducted a proteomic analysis of vacuolar membranes, isolated as described [52,53] from both wild-type and *sit4*∆ cells grown in SC medium to OD_600nm_ ≈ 2. We identified a set of 230 vacuolar membrane proteins, which represent approximately 80% of the entire vacuolar membrane proteome of *S. cerevisiae*. Proteins whose levels were altered in the *sit4*Δ vacuolar membranes are presented in Appendix A. A volcano plot (Figure 1A) was used to represent the differences in protein levels. The results reveal that 22 proteins exhibited reduced abundance whereas 27 proteins showed increased abundance (Figure 1B) in the vacuolar membranes of yeast lacking Sit4. GO analysis on biological processes revealed an enrichment in proteins related to vacuole fusion and organization, late endosome to vacuole transport, autophagy, microautophagy, polyphosphate metabolism, and metal ion homeostasis (Figure 1C and Appendix A). Among the proteins that increased the most in *sit4*∆ vacuolar membranes were Vps27 (4.1-fold) and Hse1 (2.2-fold), which form the endosomal sorting complex required for transport-0 (ESCRT-0).

The localization and levels of Vps27 are regulated by Snf1/AMPK, which promotes Vps27 translocation from endosomes to the vacuolar membranes at the PDS phase due to glucose depletion [63,64]. Notably, Snf1 is activated in *sit4*Δ cells and is critical for lifespan extension in this mutant strain [35]. To assess how *SIT4* deletion affects total Vps27 levels, *vps27*Δ and *sit4*Δ*vps27*Δ cells expressing *VPS27* fused to HA, controlled by its endogenous promotor, were grown to late Log and PDS phases, and Vps27 levels on total protein extracts were assessed by Western blotting. In agreement with the published data [63], Vps27 levels decreased in wild-type cells during growth from the Log to the PDS phase. At the Log phase, total Vps27 levels were lower in the *sit4*Δ mutant in comparison to wild-type cells (Appendix A). Overall, our results suggest that Vps27 is specifically enriched in the *sit4*Δ vacuolar membranes.

ESCRT-0 plays a key role in the late endosome to vacuole transport through the MVB sorting of ubiquitinated cargo [65,66]. In recent years, it was also implicated in the microautophagy of vacuolar membrane proteins [64]. Furthermore, a large-scale study suggests that *VPS27* deletion leads to a decrease in cell survival [67]. This led us to assess the impact of *VPS27* deletion on yeast CLS. Our results show that the loss of Vps27 had a mild negative impact on lifespan, but it abolished the lifespan extension phenotype of *sit4*Δ cells (Figure 2A,B). In fact, the longevity of *sit4*Δ*vps27*Δ cells was even lower compared to that observed in *vps27*Δ cells, indicating a strong negative genetic interaction between *SIT4* and *VPS27*. Although *sit4*Δ and *sit4*Δ*vps27*Δ cells have a slow growth phenotype (Appendix A), it is unlikely that the differences between the CLS of the different strains are related to changes in cell growth or glucose metabolism. Indeed, 24 h after Log (considered t0 in the CLS assay due to the rapid loss of viability observed in *sit4*Δ*vps27*Δ cells), all strains clearly entered the PDS phase, as shown by the growth curve, extracellular glucose (very low or absent), and ethanol (produced at similar levels) (Appendix A). Moreover, *sit4*Δ*vps27*Δ cells did not exhibit defects in the accumulation of glycogen, a storage carbohydrate that increases in *sit4*Δ cells and is critical for CLS [68].

### 3.2. The Pathways for Vacuolar Degradation Are Downregulated in *sit4*Δ Cells in a Vps27-Independent Manner

Vps27 is involved in ubiquitinated cargo recognition and subsequently in the recruitment of ESCRT-I, facilitating the sorting of ubiquitinated MVB cargo [65,66]. During nutrient depletion, e.g., during cell growth to PDS and stationary phases, the MVB pathway promotes intracellular amino acid homeostasis and the upregulation of vacuolar hydrolases [69]. In fact, the MVB pathway synergistically cooperates with autophagy to promote cell longevity following nutrient depletion [69,70]. This led us to assess if the enrichment of Vps27 in the *sit4*Δ vacuolar membranes results in MVB pathway induction.

To assess the MVB pathway, yeast cells were transformed with *MUP1-GFP*, to express a plasma membrane methionine permease that is transported for degradation inside vacuoles in response to starvation or to high levels of methionine [71]. When Mup1-GFP is degraded inside vacuoles, free GFP accumulates due to its resistance to the lytic environment of vacuoles. Therefore, the induction of the MVB pathway can be assessed by Western blotting, i.e., by measuring the percentage of Mup1-GFP degradation (free GFP/(Mup1-GFP + GFP) ratio). As expected, during the transition from the Log to PDS phase, the MVB pathway was induced in wild-type but not in *vps27*Δ cells (Figure 3A,B), as it is dependent on the ESCRT machinery [69,72]. In *sit4*Δ cells, Mup1-GFP degradation was significantly reduced at the PDS phase, indicating that MVB pathway induction was decreased. However, the addition of methionine increased Mup1-GFP degradation in *sit4*Δ cells to wild-type levels, in a Vps27-dependent manner (Appendix A), suggesting that Sit4 mediates MVB induction during cell growth but not during methionine supplementation. Overall, our data suggest that the MVB pathway does not contribute to lifespan extension in the *sit4*Δ mutant.

Nutrient depletion also induces the sorting of vacuolar membrane proteins into the lumen followed by its degradation by microautophagy, in a Vps27-dependent manner [64,73]. This process is regulated by TORC1, as it phosphorylates Vps27 to inhibit microautophagy [74], and by Snf1, as Vps27 is incapable of reaching the vacuole in cells lacking Snf1 [63]. Microautophagy can be evaluated by following the processing of GFP-tagged Pho8 or other vacuolar membrane proteins [73,74]. To monitor microautophagy, cells were transformed with *GFP*-*PHO8*, and the levels of GFP-Pho8 and GFP were assessed by immunoblotting. Consistent with published findings, microautophagy was induced in wild-type cells at the PDS phase or upon TORC1 inhibition with rapamycin (Figure 3C,D) in a Vps27-dependent manner. *SIT4* deletion significantly reduced GFP-Pho8 processing at the PDS phase, indicating that microautophagy induction was decreased in these mutant cells. However, rapamycin was capable of inducing Pho8 degradation in *sit4*Δ cells, suggesting that inhibition of microautophagy in this mutant may be caused by impaired signaling rather than defects in the machinery assisting this process. Overall, these results suggest that microautophagy does not contribute to lifespan extension in the *sit4*Δ mutant.

In the CPY pathway, the transport of proteins to vacuoles depends on multivesicular endosomes and Vps27 [75,76]. To assess if Vps27 enrichment at vacuolar membranes reflects an altered capacity of the CPY pathway in the *sit4*Δ background, CPY processing was analyzed by immunoblotting. The results showed complete processing of CPY in both growth phases in all strains (Figure 4A). However, compared with the parental strain, *vps27*Δ and *sit4*Δ*vps27*Δ cells had a tendency to display lower levels of mature CPY (mCPY) in the Log phase (Figure 4A,B). This decrease is probably due to an aberrant secretion of the precursor form of CPY (prCPY) to the extracellular space, characteristic of class E vps mutants [77]. At the PDS phase, mCPY levels in *vps27*Δ mutants were restored to wild-type levels, suggesting that other trafficking pathways may be rescuing prCPY to be processed in the vacuole. We also assessed whether deletion of *VPS27* in *sit4*Δ cells results in a CPY secretion phenotype by performing a colony immunoblot assay. As expected, *vps27*Δ cells exhibited a strong CPY secretion phenotype (Figure 4C). Notably, CPY secretion increased in the *sit4*Δ*vps27*Δ double mutant vs. wild-type and *sit4*Δ cells. These findings suggest that the CPY pathway is not altered in *sit4*Δ cells but that *VPS27* deletion impairs this pathway, which may underlie the premature aging of *sit4*Δ*vps27*Δ cells.

In addition to CPY, yeast uses the Cvt pathway, another biosynthetic trafficking pathway, to transport hydrolases, such as the Ape1, into the vacuole. For this, Cvt relies on the molecular machinery involved in autophagy [30]. To determine whether Cvt is altered in *sit4*Δ cells, we monitored Ape1 processing to its mature form. In the PDS phase, all strains exhibited near complete processing of Ape1 (Figure 5A,B). However, in the Log phase, *sit4*Δ cells had a 1.7-fold increase in Ape1 maturation compared to the parental strain. Moreover, Ape1 processing increased in the *sit4*Δ*vps27*Δ mutant vs. *vps27*Δ cells, suggesting that the deletion of *SIT4* increased the Cvt pathway in a Vps27-independent manner.

Previous studies showed that yeast ESCRT mutants exhibit autophagy defects, probably associated with a decrease in ESCRT-assisted closure of the phagophore structure, leading to the formation of the autophagic body [78]. The deletion of ESCRT-0 components (*VPS27* and *HSE1*) in the fungus *Magnaporthe oryzae* also results in abnormal autophagy [79]. Since autophagy is vital for cellular homeostasis during aging [80], we also monitored the autophagic flux by employing the GFP-Atg8 cleavage assay [81]. For this, cells were transformed with *GFP*-*ATG8* and the levels of GFP-Atg8 and GFP were monitored during cell growth (24 h and 48 h after the Log phase). As expected, the *vps27*Δ mutant exhibited a decreased autophagic flux. Furthermore, the autophagic flux was impaired in *sit4*Δ and *sit4*Δ*vps27*Δ cells (Figure 5C,D) at least up to 48 h after the Log phase. The observed inhibition of autophagy in *sit4*Δ cells appears to result from changes in signaling and not from defects in the autophagic process, since previous studies demonstrated that autophagy is induced in this mutant in response to rapamycin [33,82]. These findings suggest that the increased CLS in the *sit4*Δ mutant is unrelated to autophagy.

### 3.3. Loss of Iron Homeostasis in *sit4*∆*vps27*∆ Cells Does Not Contribute to Its Shortened Lifespan

Our proteomic analysis from *sit4*Δ vacuolar membranes also identified alterations in proteins involved in iron trafficking, namely in Smf3, Ccc1, and Fth1 (Appendix A). It is known that Vps27 impacts iron homeostasis by controlling the degradation of iron transporters. Under high iron conditions, iron transporters are sorted into vesicles by ESCRT-0 and the MVB pathway, to be degraded in vacuoles [83]. This prompted us to investigate whether *VPS27* deletion affects iron homeostasis, contributing to the shortened CLS of *sit4*Δ*vps27*Δ cells. The results showed that iron levels were similar in all strains at the Log phase (Figure 6A). However, at the PDS phase, the *sit4*Δ*vps27*Δ mutant exhibited higher levels of iron. This result led us to evaluate the sensitivity of cells to grow in medium supplemented with iron. We found that *sit4*Δ cells exhibited a higher sensitivity to iron supplementation that was aggravated by *VPS27* deletion (Figure 6B). *VPS27* deletion in *sit4*Δ cells may decrease the degradation of iron transporters, leading to the observed high iron levels and iron sensitivity. We hypothesized that iron accumulation leads to iron toxicity, contributing to the premature aging of *sit4*Δ*vps27*Δ cells. To examine this, we evaluated the impact of iron chelation with BPS, known to cause intracellular iron deprivation, on oxidative stress resistance and CLS. To assess oxidative stress resistance, cells grown in SC medium with or without BPS were treated with 1.5 mM H_2_O_2_. Our data show that *sit4*Δ*vps27*Δ cells did not exhibit increased sensitivity to H_2_O_2_ (Figure 6C). Moreover, BPS decreased the H_2_O_2_ sensitivity in all strains, but this protective effect was lower in *sit4*∆*vps27*∆ mutant cells, suggesting that iron accumulation does not aggravate H_2_O_2_ sensitivity. Moreover, BPS had a minor effect on the CLS of *sit4*Δ*vps27*Δ cells as it only increased cell survival from 1 to 2 days (Figure 6D). Overall, these findings suggest that iron accumulation does not seem to be a primary contributing factor to the drastic viability loss of *sit4*Δ*vps27*Δ cells.

### 3.4. Vps27 Is Crucial for Mitochondrial Function in *sit4*∆ Cells

Sit4 regulates mitochondrial metabolism by repressing respiratory genes in high glucose conditions [42], and mitochondrial derepression is crucial for the increased CLS of *sit4*Δ cells [32,34,35]. Sit4 modulates the catabolite repression through regulation of Mig1 and Hxk2. During cellular growth in a glucose medium, the unphosphorylated Mig1-Hxk2 complex accumulates in the nucleus, where it represses the expression of genes required for cellular growth on non-fermentable carbon sources [84,85]. In the absence of Sit4, Hxk2 is hyperphosphorylated [86] and Mig1 levels decrease [87]. Moreover, Snf1 is activated [88,89], leading to Mig1 phosphorylation and inhibition [90]. Thus, cells lacking Sit4 exhibit a high oxygen consumption rate (OCR) already at the Log phase even when grown in fermentable media [32]. A higher respiratory capacity early in growth has been related to an improved tolerance to reactive oxygen species during the stationary phase and also to an increase in lifespan [91,92]. Moreover, *sit4*Δ cells exhibit a higher dependency on the respiratory metabolism than wild-type cells [42]. These data and the lack of alterations in vacuolar trafficking pathways regulated by Vps27 that may explain its role in *sit4*Δ lifespan led us to analyze the function of Vps27 in the modulation of *sit4*Δ mitochondrial function.

To assess if Vps27 is required for the increased mitochondrial respiration displayed by *sit4*Δ cells, the OCR was measured in cells at both Log and PDS phases. Notably, *VPS27* deletion abolished mitochondrial derepression at the Log phase in cells lacking Sit4 (Figure 7A). In fact, *sit4*Δ*vps27*Δ cells had lower OCR in both growth phases, compared to wild-type and *sit4*Δ cells, indicating that the double mutant has severe deficiencies in mitochondrial respiration. Furthermore, the *vps27*Δ mutant failed to fully induce mitochondrial respiration during growth from the Log to the PDS phase and exhibited a lower OCR, compared to the parental strain. Thus, Vps27 seems to potentiate optimal mitochondrial activity, and its decrease may contribute to the shortened CLS exhibited by *vps27*Δ mutants (Figure 2). We also evaluated the ability of *sit4*Δ*vps27*Δ cells to grow in a medium containing glycerol (YPG), a respiratory carbon source, instead of glucose (YPD). The results demonstrated that *sit4*Δ*vps27*Δ cells were incapable of growing in YPG (Figure 7B), further suggesting that mitochondrial function is impaired. Overall, these results suggest that Vps27 is essential to ensure respiratory fitness and extended longevity in *sit4*Δ cells.

Numerous studies support a key role of vacuolar trafficking pathways, such as autophagy and mitophagy, in mitochondrial fitness. Mitophagy is a selective type of autophagy that targets damaged mitochondria for degradation in vacuoles. Along with mitochondrial dynamics, mitophagy is of central importance to the sustainment of mitochondrial integrity and activity [93,94,95,96]. This led us to investigate whether Vps27 is involved in the modulation of mitophagy and mitochondrial biogenesis in cells lacking Sit4. Mitophagy was assessed in cells expressing the vacuolar alkaline phosphatase Pho8 targeted to mitochondria (mtPho8) [97]. Since maturation of the Pho8 pro-enzyme requires vacuolar Pep4, the alkaline phosphatase activity is an indicator of mitochondrial degradation in vacuoles. Our results show that the alkaline phosphatase activity was higher in cells lacking Sit4 at the Log and PDS phases, although in a Vps27-independent manner (Figure 8A). In fact, we even observed higher activity in *sit4*Δ*vps27*Δ cells at the Log phase compared to *sit4*Δ cells. This indicates that the mitochondrial respiration defects in the double mutant *sit4*Δ*vps27*Δ are not associated with impaired mitophagy. Moreover, these findings suggest that enhanced mitophagy activity in *sit4*Δ cells is not associated with Vps27-dependent CLS extension.

Mitochondrial integrity depends on the coordination of both mitochondrial degradation and mitochondrial biogenesis [93,94,98]. To evaluate if *VPS27* deletion affects mitochondrial biogenesis, the levels of mitochondrial proteins (Por1, Atp2, and Cox2) were assessed in cells at the Log and PDS phases. As expected, the levels of the porin Por1 increased in wild-type cells from the Log to the PDS phase (Figure 8B,C), reflecting the higher mitochondrial mass characteristic of the respiratory phase. Accordingly with the early catabolite derepression in cells lacking Sit4, the levels of Por1 increased at the Log phase. The increase in mitochondrial biogenesis in the *sit4*Δ mutant was Vps27 independent, as *sit4*Δ*vps27*Δ also exhibited higher levels of Por1 at the Log phase. Similar changes were observed when mitochondrial mass was assessed by flow cytometry, using cells expressing preSU9-GFP (Appendix A). Notably, deletion of *VPS27* in *sit4*Δ cells abolished the higher levels of the oxidative phosphorylation proteins Atp2 and Cox2 (Figure 8B,D,E). In fact, Cox2 was not detected in *sit4*∆*vps27*∆ cells, even at the PDS phase. As Cox2 is encoded by the mitochondrial genome, we speculated that *sit4*Δ*vps27*Δ exhibits defects in the maintenance of mtDNA. However, analysis by qPCR revealed that mtDNA levels were not affected in *sit4*Δ*vps27*Δ compared to wild-type cells (Figure 8F). These observations suggest that the defects in *sit4*Δ*vps27*Δ mitochondrial function did not result from defects in mitochondrial biogenesis or the loss of mtDNA.

## 4. Discussion

The decline in the function of lysosomes (vacuoles in yeast) and mitochondria has been linked to the aging process and the development of age-related diseases [1,6]. Furthermore, numerous interventions that improve the function of these organelles also increase lifespan [7,12,13,14,15,16,17,18]. Our previous studies showed that *SIT4* deletion increases mitochondrial respiration and yeast CLS, in part mediated by increased phosphorylation levels of hexokinase 2 [86] and ATP synthase catalytic beta subunit Atp2 [34], as well as Snf1/AMPK activation [35]. The deletion of *SIT4* also suppresses the mitochondrial dysfunction exhibited by Isc1-deficient cells that accumulate ceramide during aging [32], as well as the vacuolar fragmentation and dysfunctions exhibited by this mutant [33]. However, the mechanism by which Sit4 regulates the vacuolar–mitochondrial axis is not fully understood, and its characterization may help to define new strategies to improve lifespan.

In this study, we identified 49 vacuolar membrane proteins whose levels change in cells lacking Sit4. These proteins were mainly associated with cellular processes, such as vacuole fusion and organization, late endosome to vacuole transport, autophagy, microautophagy, polyphosphate metabolism, and metal ion homeostasis. Proteins that increased significantly in *sit4*∆ vacuolar membranes include Vps27 (4.1-fold) and Hse1 (2.2-fold), components of the ESCRT-0 [65]. The enrichment of Vps27 at *sit4*∆ vacuolar membranes was not due to upregulation of Vps27 since its total protein levels were not increased. The translocation of Vps27 from endosomes to vacuolar membranes depends on Snf1 activity at the PDS phase [63] and Sit4 deficiency results in Snf1 activation early during growth in glucose medium [88,89]. Whether Snf1 activation promotes Vps27 vacuolar localization in *sit4*∆ cells remains to be clarified.

Our results suggest that Vps27 is crucial for lifespan extension in *sit4*∆ cells. Indeed, the CLS of *sit4*∆*vps27*∆ cells was much shorter than that of each single mutants, indicating a negative genetic interaction between *SIT4* and *VPS27*. The analysis of vacuolar trafficking pathways that are known to be dependent on Vps27 showed that the MVB pathway and microautophagy were impaired in both *sit4*∆ and *sit4*∆*vps27*∆ cells. However, *VPS27* deletion in *sit4*∆ cells resulted in aberrant CPY secretion. Thus, defects in the transport of cargo proteins through the CPY pathway may contribute to vacuolar dysfunction and a short lifespan in *sit4*∆*vps27*∆ cells. We also showed that the Cvt pathway was induced in both *sit4*∆ and *sit4*∆*vps27*∆ cells at the Log phase, suggesting that Vps27 does not promote *sit4*∆ CLS through regulation of Cvt. Intriguingly, the induction of autophagy at the PDS phase was impaired in *sit4*∆ and *sit4*∆*vps27*∆ cells. Autophagy induction has been largely associated with increased longevity and described as a requirement for lifespan extension by pharmacological and nutritional interventions [24,99]. However, several *atg* mutants with defective autophagy still exhibit lifespan extension under caloric restriction conditions [100]. Furthermore, several pieces of evidence suggest that autophagy and the ubiquitin proteasome system are coordinated for proper control of protein homeostasis and housekeeping functions, and that the inhibition of autophagy can be compensated by the induction of the proteasome, and vice versa [101,102,103,104]. Although Sit4 does not regulate proteasome activity, Sit4 and proteasome are functionally linked as they act in concert in osmoregulation and nutrient sensing [105]. More studies are required to assess if the ubiquitin proteasome system contributes to *sit4*∆ lifespan extension.

Our studies also suggest that Vps27 contributes to iron homeostasis, with *sit4*∆v*ps27*∆ cells accumulating high iron levels and exhibiting increased sensitivity to iron supplementation. However, iron accumulation does not seem to be involved in the *sit4*∆*vps27*∆ premature aging, as BPS treatment only promoted a minor increase in the mutant CLS. In addition, we found that Vps27 affects the levels of proteins critical for oxidative phosphorylation in *sit4*∆ cells, namely of Atp2 and Cox2. How Vps27 regulates these proteins in cells lacking Sit4 to increase mitochondrial fitness, which is critical for the *sit4*∆ mutant extended longevity, requires further investigation.

## 5. Conclusions

In summary, this study suggests that Vps27, a component of the ESCRT-0 complex, is enriched at the vacuolar membrane of *sit4*∆ cells, playing a crucial role in the CLS extension observed in this mutant strain. Interestingly, the deletion of *VPS27* in *sit4*∆ cells did not affect protein trafficking or vacuolar sorting pathways. However, it impaired mitochondrial respiration, associated with reduced levels of Atp2 and Cox2. Together, these findings unveil a novel link between Sit4, Vps27, and mitochondrial respiration, providing valuable insights into the intricate mechanisms governing cellular longevity.

## Figures and Tables

**Figure 1 cells-13-00655-f001:**
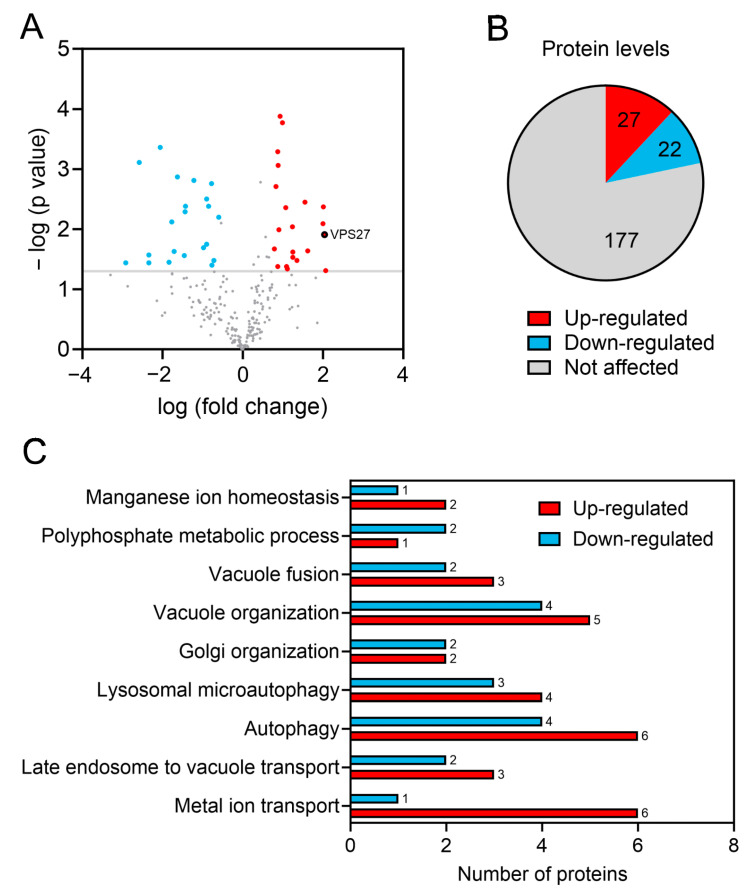
Proteomic analysis of *sit4*∆ vacuolar membranes. (**A**) Volcano plot for differentially expressed proteins in *sit4*∆ vs. wild-type samples. The log-transformed *p*-values (Student’s *t*-test) are plotted in the vertical axis against the log-transformed fold change on the horizontal axis. The horizontal dashed line indicates a *p*-value of 0.05. Red dots represent proteins that are upregulated with a 1.5-fold change threshold, whereas blue dots represent proteins that are downregulated (log2 fold change of ±0.58). Grey dots represent the unchanged proteins. Vps27 protein is identified with a red dot with a black outline. (**B**) Number of proteins significantly up- or downregulated in the *sit4*∆ vacuolar membranes. (**C**) GO term enrichment analysis on biological processes was conducted for proteins exhibiting statistical alterations in *sit4*∆ samples.

**Figure 2 cells-13-00655-f002:**
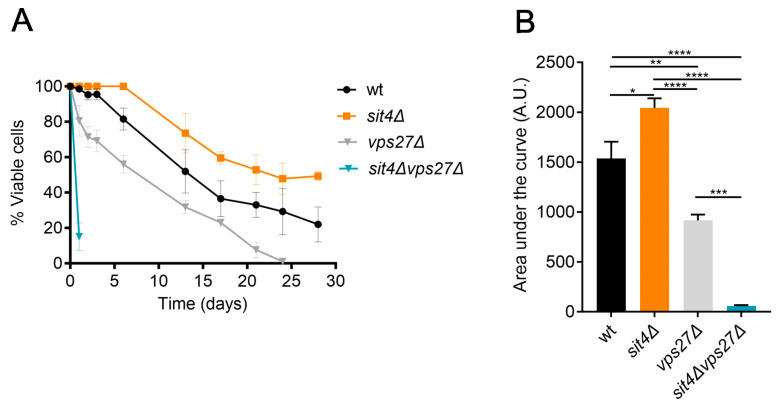
*SIT4* deletion extends lifespan in a Vps27-dependent manner. (**A**) Cells were grown to PDS phase (24 h after Log; t0) and maintained over time in SC medium. Cellular viability was expressed as the percentage of CFUs (aged vs. t0). Data are the mean ± SEM (*n* ≥ 3). (**B**) The area under each lifespan curve was calculated using GraphPad and expressed in arbitrary units (A.U.). Data are the mean ± SEM (*n* ≥ 3); * *p* ≤ 0.05; ** *p* ≤ 0.01; *** *p* ≤ 0.001; ****, *p* ≤ 0.0001; one-way ANOVA.

**Figure 3 cells-13-00655-f003:**
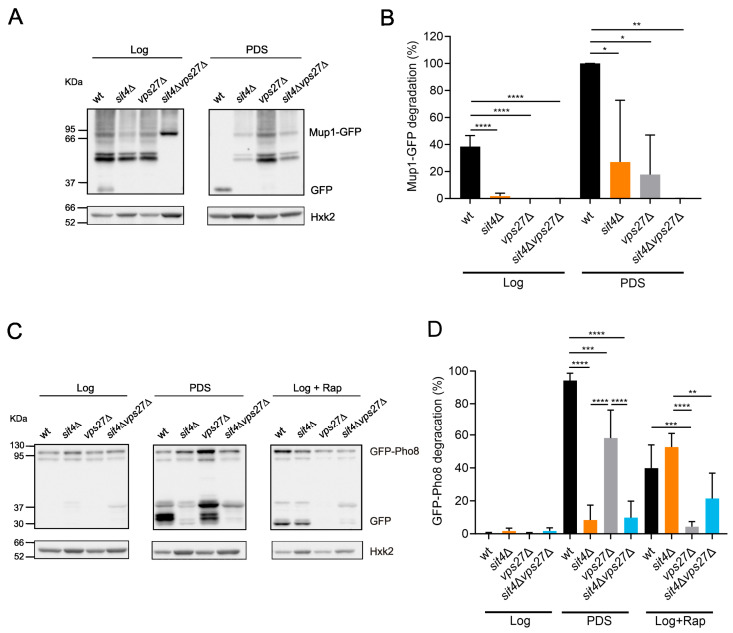
The induction of the MVB pathway and microautophagy at the PDS phase is Sit4-dependent. (**A**) Immunodetection of Mup1-GFP and GFP in protein extracts from cells expressing pRS416-*MUP1*-*GFP* grown in SC medium. Hxk2 was used as a loading control. A representative blot is shown. (**B**) The induction of the MVB pathway (free GFP/(Mup1-GFP + GFP) ratio) was evaluated at the Log and PDS (24 h after Log) phases. Values are the mean ± SD (*n* = 3); * *p* < 0.05, ** *p* < 0.01, **** *p* < 0.0001; one-way ANOVA. (**C**) Immunodetection of GFP-Pho8 and GFP in protein extracts from cells expressing pRS426-*GFP*-*PHO8*. Hxk2 was used as a loading control. A representative blot is shown. (**D**) The induction of microautophagy (free GFP/(GFP + GFP-Pho8) ratio) was evaluated at the Log and PDS (24 h after Log) phases. Cells treated with rapamycin were used as a positive control. Values are the mean ± SD (*n* = 3); ** *p* < 0.01, *** *p* < 0.001, **** *p* < 0.0001; one-way ANOVA.

**Figure 4 cells-13-00655-f004:**
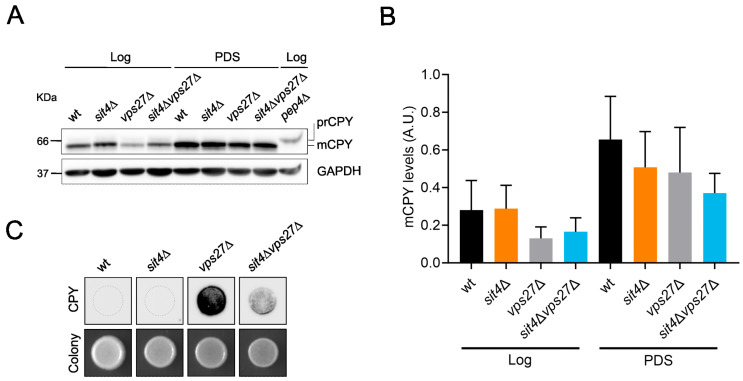
*VPS27* deletion impairs the CPY pathway in the *sit4*Δ mutant. (**A**) Immunoblotting showing complete processing of CPY in both Log and PDS (24 h after Log) phases in all strains (except in *pep4*Δ cells, used as control). (**B**) CPY protein levels were quantified using GAPDH as a loading control. Values are the mean ± SD (*n* = 4). (**C**) CPY secretion was assessed using a colony immunoblot assay. A representative image is shown (*n* = 3).

**Figure 5 cells-13-00655-f005:**
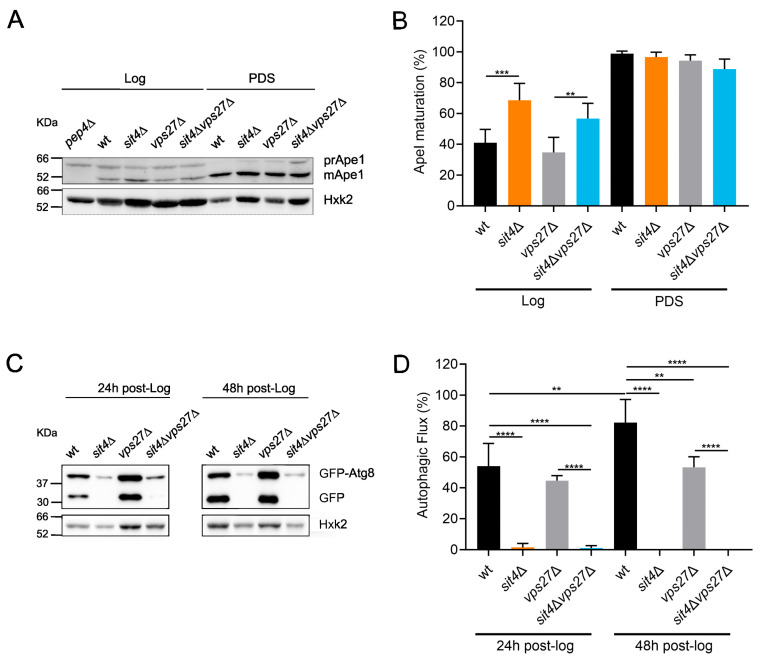
*SIT4* deletion compromises autophagy and induces the Cvt pathway at the Log phase in a Vps27-independent manner. (**A**) Processing of the proenzyme ApeI (prApeI) to the mature enzyme (mApeI) was analyzed at the Log and PDS (24 h after Log) phases. Hxk2 was used as a loading control. The *pep4*∆ cells (unable to process ApeI) were used as control. A representative blot is shown. (**B**) Quantification of ApeI maturation. Values are the mean ± SD (*n* = 4); ** *p* < 0.01, *** *p* < 0.001; one-way ANOVA. (**C**) Immunodetection of GFP-Atg8 and GFP in cells expressing pRS416-*GFP*-*ATG8*. Hxk2 was used as a loading control. A representative blot is shown. (**D**) The autophagic flux (free GFP/(GFP + GFP-Atg8) ratio) was evaluated at 24 h and 48 h after the Log phase. Values are the mean ± SD (*n* = 3); ** *p* < 0.01, **** *p* < 0.0001; one-way ANOVA.

**Figure 6 cells-13-00655-f006:**
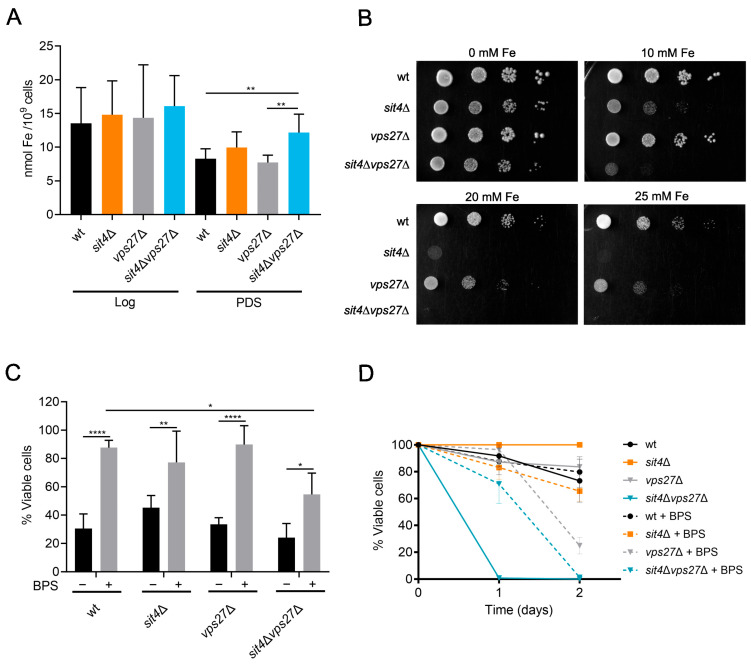
Iron levels and the effect of iron chelation on H_2_O_2_ resistance and CLS in *sit4*∆*vps27*∆ cells. (**A**) Total iron levels were quantified in cells grown to late Log and PDS (24 h after Log) phases. Values are the mean ± SD (*n* ≥ 6); ** *p* ≤ 0.01; one-way ANOVA. (**B**) Cells were grown to the Log phase and four ten-fold serial dilutions were spotted onto SC plates with or without supplementation of iron (II) sulphate heptahydrate. A representative image is shown (*n* = 3). (**C**) Cells were grown to Log phase in medium with or without supplementation of 80 µM of BPS and treated with 1.5 mM H_2_O_2_ for 1 h. Cell viability was expressed as the percentage of the CFUs (treated cells vs. non-stressed cells). Values are the mean ± SD (*n* ≥ 3); * *p* ≤ 0.05, ** *p* ≤ 0.01, **** *p* ≤ 0.0001; one-way ANOVA. (**D**) Cells were grown to the PDS phase (t0) in SC medium with or without supplementation of 80 µM BPS and maintained over time in the same medium. Cellular viability was quantified as the percentage of the CFUs (aged vs. t0). Data are the mean ± SEM (*n* ≥ 3).

**Figure 7 cells-13-00655-f007:**
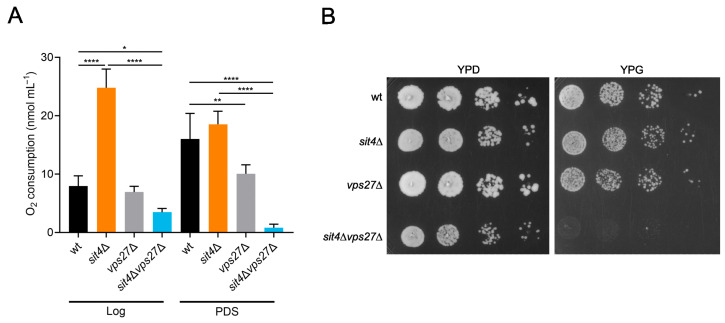
The *sit4*Δ*vps27*Δ mutant has impaired mitochondrial respiration. (**A**) Cells were grown to Log and PDS phases and mitochondrial respiration was determined by measuring oxygen consumption rate. Values are the mean ± SD (*n* ≥ 4); * *p* < 0.05, ** *p* < 0.01, **** *p* < 0.0001; one-way ANOVA. (**B**) Cells were grown to Log phase and four ten-fold serial dilutions were spotted onto YPD or YPG plates. A representative image is shown (*n* = 4).

**Figure 8 cells-13-00655-f008:**
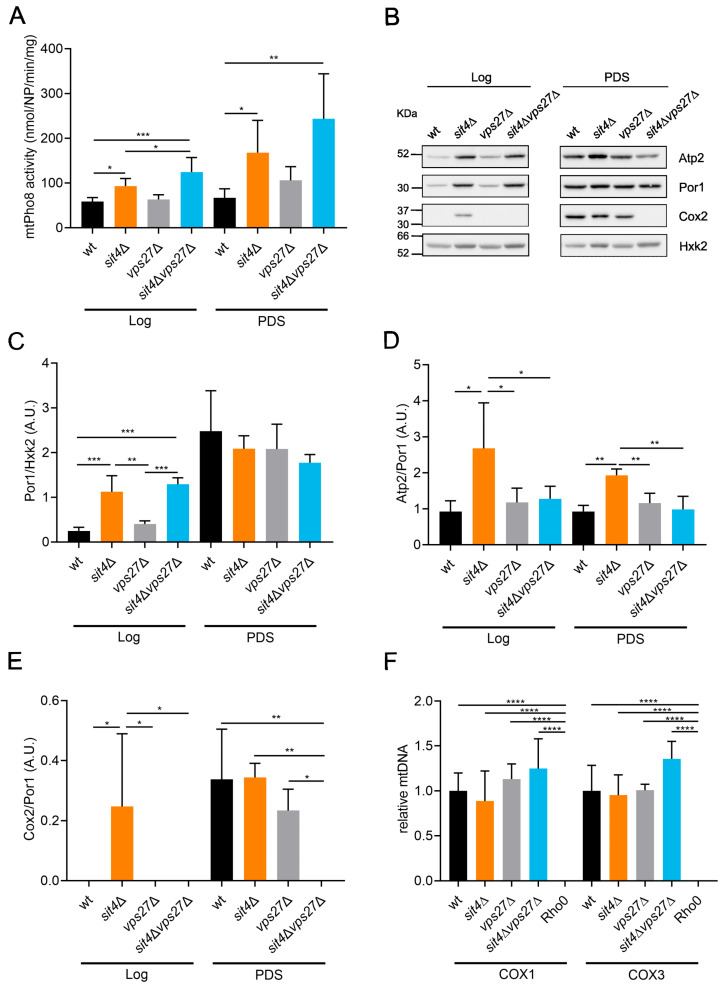
*VPS27* deletion in *sit4*∆ cells leads to a decrease in Atp2 and Cox2 levels, but it does not affect mitophagy or mitochondrial biogenesis. (**A**) Mitophagy is induced in *sit4*Δ cells in a Vps27-independent manner. Cells expressing pYX242-*mtPHO8* were grown to Log and PDS (24 h after Log) phases and mitophagy was determined by measuring the alkaline phosphatase activity. Values are the mean ± SD (*n* = 4); * *p* ≤ 0,05, ** *p* ≤ 0.01, *** *p* < 0.001; one-way ANOVA. (**B**) Cells were grown to Log and PDS phases and the levels of mitochondrial proteins (Por1, Atp2, and Cox2) were assessed by immunoblotting. Hxk2 was used as a loading control. A representative blot is shown (*n* = 3). (**C**) Relative amount of Por1 (a mitochondrial outer membrane protein) to Hxk2 is shown as an indicator of mitochondrial mass. Values are the mean ± SD; ** *p* ≤ 0.01, *** *p* < 0.001; one-way ANOVA. (**D**,**E**) Quantification of the ratio Atp2/Por1 (**D**) and Cox2/Por1 (**E**). Values are the mean ± SD; * *p* ≤ 0.05, ** *p* ≤ 0.01; one-way ANOVA. (**F**) Analysis of mtDNA/nuclear DNA by qPCR. *COX1* and *COX3* were used as reference for mtDNA and *ACT1* for nuclear DNA. Rho0 cells were used as control. Values are the mean ± SD (*n* = 4); **** *p* < 0.0001; one-way ANOVA.

**Table 1 cells-13-00655-t001:** *Saccharomyces cerevisiae* strains used in this study.

Strain	Genotype	Source
BY4741 ^b,e,f^	Mata, *his3*Δ1, *leu2*Δ0, *met15*Δ0, *ura3*Δ0	EUROSCARF
*sit4*Δ ^b,e,f^	BY4741 *sit4*::*MXHIS3*	[34]
*vps27*∆ ^a,b,e,f^	BY4741 *vps27*::*KanMX4*	This study
*sit4*∆*vps27*∆ ^a,b,e,f^	BY4741 *sit4*::*MXHIS3 vps27*::*KanMX4*	This study
*pep4*Δ	BY4741 *pep4*::*KanMX4*	EUROSCARF
*pho8*Δ ^c^	BY4741 *pho8*::*HPH*	[49]
*sit4*Δ*pho8*Δ ^c^	BY4741 *sit4*::*KanMX4 pho8*::*HPH*	[49]
*vps27*Δ*pho8*Δ ^c^	BY4741 *vps27*::*MXHIS3 pho8*::*HPH*	This study
*sit4*Δ*vps27*Δ*pho8*Δ ^c^	BY4741 *sit4*::*KanMX4 vps27*::*MXHIS3 pho8*::*HPH*	This study
BY4742 ^d^	Matα *his3*Δ1, *leu2*Δ0, *lys2*Δ0, *ura3*Δ0	EUROSCARF
*sit4*Δ ^d^	BY4742 *sit4*::*KanMX4*	EUROSCARF
*vps27*∆ ^d^	BY4742 *vps27*::*MXHIS3*	This study
*sit4*∆*vps27*∆ ^d^	BY4742 *sit4*::*KanMX4 vps27*::*MXHIS3*	This study
BY4741 Rho0	BY4741 Rho0	[50]

Cells harboring ^a^ YCpHAC33-*Prom*-*VPS27*-*3xHA*, ^b^ pRS416-*GFP-ATG8*, ^c^ pYX242-*mtPho,*
^d^ pRS416-*MUP1*-*GFP*, ^e^ pRS426-*GFP*-*PHO8*, and ^f^ pVT100U-*preSU9-GFP* are indicated.

**Table 2 cells-13-00655-t002:** Sequences of primers used in qPCR.

Primer	Sequence
COX1 fw	CTACAGATACAGCATTTCCAAGA
COX1 rv	GTGCCTGAATAGATGATAATGGT
COX3 fw	TTGAAGCTGTACAACCTACC
COX3 rv	CCTGCGATTAAGGCATGATG
ACT1 fw	GTATGTGTAAAGCCGGTTTTG
ACT1 rv	CATGATACCTTGGTGTCTTGG

## Data Availability

The mass spectrometry proteomics data have been deposited to the ProteomeXchange Consortium via the PRIDE [106] partner repository with the dataset identifier PXD048450.

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
