# Peer review of "Sit4 Genetically Interacts with Vps27 to Regulate Mitochondrial Function and Lifespan in Saccharomyces cerevisiae"

_cells, 2024, doi:10.3390/cells13080655_

Round 1

Reviewer 1 Report

Comments and Suggestions for Authors

Martins T.S. et al. “Sit4 genetically interacts with Vps27 to regulate mitochondrial function and lifespan in Saccharomyces cerevisiae” manuscript submitted to Cells.

This work of Martins and colleagues extends and completes previous studies of the group already published that concern Sit4, a phosphatase involved in many processes including yeast chronological aging. In particular, here, they analyzed the effects of the lack of Sit4 on vacuolar and mitochondrial functions concluding that Vps27 is crucial for the CLS extension that characterizes sit4 null mutants through regulation of mitochondrial function.

The paper is well organized and of interest. All the results are presented clearly and discussed. The study is also properly referenced. Although I think that the data presented contain enough novelty and the manuscript deserves to be published on Cells, I have just a few comments that should help to clarify some aspects.

Below please find some points requiring further elaboration or revision by the authors:

Since the paper focused on the behavior of the different null mutants throughout the yeast’s entire chronological life span, the experimental set up should be better defined.

I suggest to include a supplemental figure showing some parameters such as cell number, extracellular glucose consumption and ethanol production of the different cultures (sit4, vps27 and sit4vps27 null mutants). In such a way it is possible to define the growth profile (exponential phase, diauxic shift, post-diauxic phase and stationary phase) of the different cultures and consequently the metabolic state during which the different analyses have been performed (not only “ overnight cultures were diluted….and after 24h or 48h).

I think that a clear description of the metabolic state can be helpful to understand changes of the different strains/conditions.

In this context:

the growth rates of the different cultures were the same? To the best of my knowledge the sit4 null mutant has a slow growth phenotype and this is not the case for the vps27 null mutant. And the double mutant?

 and it is just a curiosity: why cells were grown at the temperature of 26 °C and not at 30 °C? BY4741 and BY4742 strains are not ts mutants?

In addition, during chronological aging, after the diauxic shift, a metabolic change from fermentation to respiration takes place implying that energy metabolism relies on mitochondrial functionality. The sit4 vps27 null mutant that according to the results displays defects in the maintenance of mitochondrial DNA can perform the diauxic shift and use ethanol, a C2 compound, previously produced? Consequently no gluconeogenesis and glucose stores? Premature aging/no viability due to this?

Concerning mitochondrial respiration authors measured total oxygen consumption in the different mutants. Since carbon starvation elicits a transition from phosphorylating to non-phosphorylating respiration and coupling of respiration with ATP synthesis has a great influence on CLS, it will be interesting to determine other respiratory parameters such as non-phosphorylating respiration and maximal respiratory capacity in order to have a more complete characterization of the respiratory efficiency of the different mutants and the outcomes on CLS.                                       

Minor editing of English language is required. For example

Line 60: which mediateà mediates

Line 90: Sit4-associated proteins,......bindsà bind

Line 251: modulateà modulates

Line 259: proteins levels à protein levels

Minor points:

Lines 196, 197, 207, 212: 26°C à 26 °C

Line 325: and3Bà and 3B

Line 355: Pho8-GFPà GFP-Pho8

Comments on the Quality of English Language

Minor editing of English language is required. For example

Line 60: which mediateà mediates

Line 90: Sit4-associated proteins,......bindsà bind

Line 251: modulateà modulates

Line 259: proteins levels à protein levels

Reviewer 2 Report

Comments and Suggestions for Authors

In the manuscript titled “Sit4 genetically interacts with Vps27 to regulate mitochondrial function and lifespan in Saccharomyces cerevisiae” the authors have performed a proteomic analysis of isolated vacuolar membrane and used yeast genetic interaction analysis to show that SIT4 exhibits negative genetic interaction with VSP27. Importantly, the authors convincingly show that VPS27 is crucial for lifespan extension in sit4Δ cells, however, how VPS27 affects this process was not clear, although authors screened for different pathways that may be involved. In summary, this is a detailed work showing a novel link between Sit4, Vps27, and mitochondrial function.

Major comments:

1. Authors have shown that VPS27 abundance increases on the vacuolar membrane in sit4Δ yeast mutants using proteomic analysis. Since this is the basis of their story, they should confirm this using Western blotting analysis of the vacuolar membrane. Authors already have a vps27Δ+ VPS27-HA it would be an easy experiment to do. This experiment is also important because, in Fig. S1A, Vps27 levels decreased in the cells grown to the exponential phase as compared to WT, which may be confusing to readers.

2. In lines 533-534 authors state “Using live cells stained with DAPI and visualized by fluorescence microscopy, we were not able to detect mtDNA in sit4Δvps27Δ mutant cells (Figure 8F).” The authors can confirm their hypothesis by doing qPCR to measure mtDNA.

Minor comments:

1. Provide molecular weight markers in the western blot images.

2. In Fig.S2A, the cleaved GFP increases over time after methionine addition in WT and sit4Δ yeast cells. Supporting this observation the levels of Mup1-GFP are decreased in sit4Δ cells over time. However, it appears from the western blot data that Mup1-GFP levels do not change in the WT cells even after increased GFP cleavage. The authors should provide a reason for this in the result or discussion section.

Reviewer 3 Report

Comments and Suggestions for Authors

Martins et al. have extensively examined the genetic interactions between Sit4 and Vps27, elucidating potential pathways governing the regulation of mitochondrial function and lifespan in Saccharomyces cerevisiae. The authors employed a proteomics approach to specifically identify the involvement of Vps27 in cells lacking Sit4 (Δsit4). Despite the commendable significance of this study in advancing our understanding of these molecular interactions, I believe there are several concerns that warrant attention and clarification.

Comment 1:

The manuscript consistently employs a vector expression system (YCpHAC33-Prom-VPS27-3HA) to assess the steady-state levels of the Vps27 protein in both Δsit4 and Δsit4Δvps27 cells. Despite the control of expression by the endogenous promoter, it would have been more suitable to utilize genomic DNA tagging, particularly when quantifying steady-state protein levels within the cells.

Comment 2:

In Figures 2A and 6D, the lifespan of strains was assessed post the diauxic shift (PDS) phase. Nevertheless, it would be valuable to explore the impact of these deletion strains on the regular chronological lifespan and compare it with the observations during the PDS phase. This broader analysis is essential as alterations in lifespan observed in the deletion strains might not be confined solely to the post-diauxic phase. To substantiate their claims, the authors could provide additional evidence by investigating and presenting data on the overall chronological lifespan affected by the deletion strains, offering a more comprehensive understanding of the observed effects.

Comment 3:

In Figure 8, the ratio of Por1 to Hxk2 may not serve as a reliable indicator for mitochondrial mass. Notably, various isomers of Hxk2 are acknowledged to be localized within mitochondria, in addition to other cellular organelles.

“The Plant Cell, Volume 18, Issue 9, September 2006, Pages 2341–2355, https://doi.org/10.1105/tpc.106.041509”

To enhance the validity of these findings, it is recommended to corroborate the results using established mitochondrial mass indicators, such as staining with NAO or TMRE, followed by quantitative assessment through flow cytometry. This approach would provide a more robust and conventional measure of mitochondrial mass, ensuring the accuracy and reliability of the reported observations.

Comment 4:

In Figure 8, the assertion regarding mitochondrial DNA loss in the Δvps27 strain, as determined through DAPI staining, appears to be somewhat preliminary for drawing definitive conclusions. It is advisable to substantiate this claim by conducting additional validation, such as employing q-RT-PCR or semi-quantitative PCR targeting specific mitochondrial-encoded genes. A comparative analysis with corresponding nuclear-encoded genes would provide a more robust and thorough assessment, allowing for a more refined understanding of the observed effects on mitochondrial DNA in the Δvps27 strain.

Comment 5:

In the abstract (line no. 20) the gene Vps27 is mentioned as ‘VSP27’ – correct this accordingly.

Comment 6:

BPS was used in the experiments. It must be abbreviated, clearly mentioned in the manuscript what is the exact function of BPS and the rationale behind the treatment.

Comments on the Quality of English Language

Paper is well written except few minor corrections which are mentioned in the comments. Overall it would be better to tone down the conclusions a bit. 

Round 2

Reviewer 1 Report

Comments and Suggestions for Authors

Authors responded satisfactorily to comments/requests

Reviewer 2 Report

Comments and Suggestions for Authors

The authors have carefully considered and responded to the critiques from the previous review. I have no further suggestions for the authors.

Reviewer 3 Report

Comments and Suggestions for Authors

I appreciate the authors for addressing most of my concerns in the revised manuscript.